# Why Do They Leave? The Counterplans to Continue Working among Preschool Workers in Japan: A Cross-Sectional Survey

**DOI:** 10.3390/children10010032

**Published:** 2022-12-24

**Authors:** Moemi Matsuo, Misako Higashijima

**Affiliations:** Rehabilitation Sciences, Nishi Kyushu University, Kanzaki 842-8585, Japan

**Keywords:** mental health, preschool workers, turnover rate, well-being, willingness to continue working

## Abstract

Three institutions predominantly care for preschool children in Japan: kindergartens, authorized childcare institutions, and nursery centers. Recently, the turnover rates of preschool workers in these institutions have been high, and Japan has been facing a shortage of kindergarten teachers. The study aimed to provide concrete counterplans to reduce preschool workers’ turnover rate. To determine the causes of turnover, we conducted a cross-sectional survey. We recruited preschool workers from several kindergartens, authorized childcare institutions, and nursery centers in Japan to fill out a survey regarding counterplans for employment. Of the 1002 surveys, 551 (541 women; 10 men) complete surveys were received (response rate: 55%). A total of 295 participants answered that they were unwilling to continue working for longer than five years and completed the questionnaires. The Jiro Kawakita method was used to categorize and analyze the four sections of the counterplan questionnaires. The results showed that the main reasons for high turnover were overtime work, low salary, and difficult human relations. To solve these issues, the counterplan ideas such as workshop ideas and conditions conducive to continuing working longer were related to human relations, work conditions, and mental health.

## 1. Introduction

The three institutions that care for preschool children in Japan are kindergartens (3–5 years old), authorized childcare institutions (from birth to 5 years old), and nursery centers (from birth to 5 years old). According to the Cabinet Office of Japan, kindergarten is defined as a preschool that offers education-based playing and singing, practical activities such as drawing, and social interaction as part of the transition from home to school; it is an ambulatory institution that cares for babies and toddlers whose guardians are working. An authorized childcare institution is a center that has both kindergarten and nursery center functions. Recently, the turnover rates of preschool workers in these institutions have been high [1,2]. The importance of reducing the turnover rate among preschool workers is increasing in developed countries, including those in East Asia [3], and Japan has been facing a shortage of kindergarten teachers. These workforce changes have resulted in insufficient childcare in these institutions [4].

Researchers have attempted to solve this problem by identifying individual and environmental factors that correlate with preschool workers’ willingness to continue working. In particular, a previous study found that sex, age, mental health, social support, and work engagement were associated with teachers’ willingness to continue working [4]. Another study suggested that age, family environment, work responsibilities, mental health, and work engagement were significantly associated with the willingness to continue working. Accordingly, welfare benefits and individual support systems can be key elements in encouraging teachers to continue working and improve their job satisfaction, mental health, and wellbeing. In addition, balanced work conditions and workers’ high agreement with workplace childcare/education policies may reduce turnover [5].

Although earlier studies have provided suggestions for reducing the turnover rate, more concrete counterplans should be investigated to implement them. Thus, the main objective of this study was to determine the reasons preschool/nursery center/kindergarten teachers and workers leave employment and to provide concrete counterplans to reduce the turnover rate in these occupations. It aimed to find counterplans to reduce preschool workers’ turnover rate and contribute to the social issues in Japan.

## 2. Materials and Methods

### 2.1. Participants

This study recruited 1002 preschool workers in Japan as its potential participants. Of them, 451 participants either did not fully answer the questionnaires or did not return them. Thus, 551 complete surveys were received (response rate: 55%). Finally, of the 551 potential participants, 295 answered that they were unwilling to continue working for longer than five years and completed the questionnaires.

Informed consent was obtained from all participants prior to the study. The study was approved by the Ethics Committee of Nishi Kyushu University (approval No. 21VDV15) and complied with the Declaration of Helsinki [6].

### 2.2. Procedure

The study was conducted in 2018 as a cross-sectional survey. We recruited participants from kindergartens, authorized childcare institutions, and nursery centers in representative cities in Japan. We included only full-time teachers and individuals working in non-managerial positions to ensure that part-time workers with fixed-term contracts and managers would not affect the results. The study questionnaires took approximately ten minutes for participants to complete. All participants provided written informed consent in their workplaces and completed the survey about counterplans that could reduce the turnover rate. All questionnaire responses were self-reported and anonymous, and participants returned the completed questionnaires in sealed envelopes.

### 2.3. Measures

First, the participants were asked to choose the reasons for their unwillingness to continue working for longer than five years. Second, they were asked about their motivations for leaving and counterplan ideas. The counterplans questionnaire comprised four sections: reasons for considering leaving employment, contents of overtime work, necessary workshops to solve work problems, and ideas for continuing to work longer. These questions were all answered descriptively.

### 2.4. Data Analysis

All the data collected in this study were descriptive data. Thus, data categorization was required to calculate the result. To this end, four sections of the counterplans questionnaire described in the Measures section (above) were categorized using the Jiro Kawakita method (KJ method), also known as affinity diagramming [7]. The KJ method is widely used in participatory learning as a means to collect and organize information. It extracts and categorizes key words from the results of the questionnaire using text mining.

## 3. Results

### 3.1. Willingness to Participate

The reasons for the unwillingness to continue working for longer than five years are shown in Table 1. The main factor was enormous overtime work, followed by low salaries and difficult human relations. The results of the counterplans questionnaire are shown in Table 2, Table 3, Table 4 and Table 5. Table 2 presents the reasons for participants’ willingness to quit a job, with most responses attributing “low salary” for it, followed by “too much overtime work/no payment for overtime” and “human relations”. Table 3 describes the content of overtime work at the workplace and at home. At the workplace, “preparation for events” received the most responses, followed by “administrative work” and “extension of childcare/education”. Most participants chose “administrative work” for the content at home, followed by “preparation for events” and “making teaching materials”.

### 3.2. Workshop Idea Categories

The workshop ideas were divided into the following eight sections by types of ideas: “children’s guardians”, “taking care of children”, “understanding children”, “place to share suffering”, “basic skills for work”, “mental health”, “the way of working”, and “others.” The ideas to solve job problems with the highest percentage were “the ways to take care of/support children’s guardians”, followed by “the ways to take care of children requiring support” and “create places to share suffering and advisement” (Table 4). The ideas regarding conditions conducive to continue working longer were divided into the following 12 sections based on types of ideas: “employment conditions”, “workforce”, “workload”, “human relations”, “relationship with boss/seniors”, “work environment”, “welfare benefits”, “duty framework”, “taking care of children”, “taking care of children’s guardians”, “mental health”, and “others.” The ideas with the highest percentages were “better salary”, followed by “more workers”, and “easier to have holidays/private time” (Table 5).

**Table 4 children-10-00032-t004:** Categorized workshop ideas to solve full-time preschool workers’ job problems.

Rank No.	Workshop Ideas for Proposed Solutions	*n* (%) *n* = 479
Guardians		
1	The ways to take care of/support children’s guardians	41 (9)
15	Common understanding between guardians and childcare workers	18 (4)
24	Workshop for children’s guardians	11 (2)
38	The ways to support parents and children	7 (1)
Taking care of children		
2	The ways to take care of children requiring support	39 (8)
7	The ways to take care of children with developmental disorder	27 (6)
12	The ways to take care of children	21 (4)
19	Methods of activities for childcare	13 (3)
18	Case examination of children	14 (3)
21	Highly professional workshop	12 (3)
25	Contents of childcare/the ways of childcare	11 (2)
31	The ways to take care of children in the grey zone/undiagnosed	9 (2)
32	Methods of playing and hand playing	9 (2)
34	Practical workshop/experience-based workshop	9 (2)
40	Skill up	7 (1)
Understanding children		
29	Group discussion	10 (2)
41	The ways to understand children	6 (1)
42	The ways to understand disability	6 (1)
43	Special needs of children	6 (1)
Place to share suffering		
3	Create places to share suffering and advisement	35 (7)
4	Interaction with other kindergartens and workers working in different places	34 (7)
36	Interaction with coeval workers	9 (2)
37	Interaction with childminders	8 (2)
Basic skills for work		
8	Human relations	26 (5)
11	Communication methods	23 (5)
28	Methods of coaching freshers and juniors	10 (2)
47	Common sense/manners	5 (1)
Mental health		
13	Refresh for workers’ mind	21 (4)
16	Resolution of stress for workers	15 (3)
17	Mental health for workers	14 (3)
20	Exercises/dance for workers	13 (3)
27	Psychology	10 (2)
30	Counseling	10 (2)
45	The ways to think	6 (1)
46	Relaxation	6 (1)
The ways of working		
9	Workshop for directors and bosses	26 (5)
14	Methods for efficient work	18 (4)
22	High quality workshop inside of kindergarten	12 (3)
23	The ways to work/work system	11 (2)
26	The ways of class governance	10 (2)
33	How to work	9 (2)
35	All-hands workshop	9 (2)
44	Cooperative framework	6 (1)
Others		
5	Workshop is burden	30 (6)
6	The problems cannot be solved with workshops	29 (6)
10	Others	24 (5)
39	Workshop is unnecessary	7 (1)

The numbers on the left indicate the ideas’ ascending ranking by n (%). The ideas were divided into eight sections based on the types of ideas. The workshop ideas to solve job problems with the highest percentages were “the ways to take care of/support guardians”, followed by “the ways to take care of children requiring support” and “create places to share suffering and advisement”.

**Table 5 children-10-00032-t005:** Categorized ideas of effective conditions for preschool workers to stay longer in the job.

No.	Workshop Ideas to Solve Job Problems/Improve Conditions	*n* = 716 n (%)
Employment conditions		
1	Better salary	310 (43)
3	Easier to have holidays/private time	83 (12)
6	Overtime work is paid	57 (8)
10	Easier to have paid holidays	44 (6)
11	Can leave work on time	37 (5)
23	Two days’ holiday in a week	16 (2)
49	Have holidays on weekdays	7 (1)
50	Increase in paid holidays	7 (1)
58	Can have long vacation	6 (1)
59	No work on days off	6 (1)
Workforce		
2	More workers	95 (13)
18	More subsidiary/workers for miscellaneous duties	26 (4)
61	More full-time workers	6 (1)
Workload		
4	Less or simplification of administrative work/document work	81 (11)
5	Less take-out work	76 (11)
7	Less/simplification of kindergarten festivals	48 (7)
8	Less/no overtime work	47 (7)
9	Less workload	46 (6)
12	Can do document work in work time	35 (5)
20	Effective and equal assigning tasks	20 (3)
34	Less workshops	10 (1)
38	Less/simplification of making materials/preparation of childcare	9 (1)
69	Can prepare for childcare in work time	5 (1)
Human relations		
13	Good human relations	32 (4)
30	Atmosphere to enable advice seeking	13 (2)
32	Better communication with guardians/children/colleagues	12 (2)
41	A friendly workplace	9 (1)
42	Place to share suffering	9 (1)
45	Agreement among coworkers	8 (1)
54	Discussion between workers	7 (1)
56	Teamwork of workers	7 (1)
72	Common understanding of children between workers	5 (1)
Relationship with boss/seniors		
25	Directors and boss understand the field	15 (2)
39	Good relationship with boss	9 (1)
44	Boss listens to opinions of workers	8 (1)
52	Evaluation and praise by directors and boss	7 (1)
57	No power harassment	7 (1)
65	Can tell opinions and discuss with not only juniors but also seniors	6 (1)
71	Boss listens to workers’ suffering and takes care of it	5 (1)
Work environment		
14	Governance of duty hours	32 (4)
15	Readjustment of work framework/work contents	29 (4)
16	Readjustment of staffing standards	28 (4)
17	Can have break time	27 (4)
37	Less burden on the ones in charge	9 (1)
47	Good workplace atmosphere	8 (1)
51	Efficient work	7 (1)
60	Use of timecard	6 (1)
62	Digitalization/use artificial intelligence	6 (1)
67	Fun workplace	6 (1)
74	Management well	5 (1)
Welfare benefits		
19	Can balance work with family life/marriage and giving birth	26 (4)
21	Better welfare (allowance of house/sustenance/commutation/maternal leave, etc.)	19 (2)
26	Better compensation package for workers	14 (2)
Duty framework		
22	Abrogation of one person being in charge/Having multiple persons in charge	17 (2)
24	Have cooperative framework	16 (2)
40	Small class	9 (1)
63	Concert of policies/restructuring of workplace	6 (1)
Taking care of children		
29	Have specialists for special needs/visits by advisors	13 (2)
46	Acceptance of and environment for handicapped children	8 (1)
66	Place and time to take care of guardians and children enough	6 (1)
68	Skill-up/improvement of childcare quality	6 (1)
70	Sufficient workshops	5 (1)
73	Improvement of facility and environment for childcare	5 (1)
Taking care of guardians		
31	Check actual needs of childcare service (guardians’ days off/weekends/extended childcare)	12 (2)
35	Better understanding of kindergarten policy by guardians	10 (1)
43	Understanding of expertness and evaluation by guardians and society	9 (1)
48	Less taking care of guardians	8 (1)
64	Place to make guardians understand about children (developmental disorders, etc.)	6 (1)
Mental health conditions		
36	Good mental health	10 (1)
53	Counselor	7 (1)
55	Work engagement	7 (1)
Others		
27	Others	14 (2)
28	Improvement of the way to train up freshers	13 (2)
33	Advancement in social status of kindergarten teachers	12 (2)

The numbers on the left indicate the ideas’ ascending ranking by n (%). The ideas were divided into 12 sections based on the type of ideas. The ideas of effective conditions to continue working longer with the highest percentages were “better salary”, followed by “more workers”, and “easier to have holidays/private time”.

## 4. Discussion

This study investigated the reasons for unwillingness to continue working for longer than five years to provide concrete ideas to reduce the turnover rate among preschool teachers in kindergartens, authorized childcare institutions, and nursery centers. The primary factors for unwillingness to continue working for longer than five years were overtime work, followed by low salary and human relations, which were consistent with the findings of previous research [5]. Additionally, the reasons for wanting to quit a job with the highest percentages were low salary, followed by too much overtime work/no payment for overtime and human relations, which were similar to the factors for unwillingness to continue working for longer than five years [5]. Preparation for events was indicated as the content at workplace with the highest percentage of responses, followed by administrative work and the extension of childcare/education. The contents at home with the highest percentages were administrative work, followed by preparation for events and making teaching materials. These results indicate that workers may have high levels of stress resulting from the need to work a lot, even in their private time, with low salaries in workplaces with difficult human relations.

To solve these issues, we asked the participants about their counterplan ideas. The results showed that the workshop ideas to solve job problems were related to “children’s guardians”, “taking care of children”, “understanding children”, “place to share suffering”, “basic skills for work”, “mental health”, and “the way of working.” In addition, the ideas of conditions most conducive to continue working longer were related to “employment conditions”, “workforce”, “workload”, “human relations”, “relationship with boss/seniors”, “work environment”, “welfare benefits”, “duty framework”, “taking care of children”, “taking care of children’s guardians”, and “mental conditions.”

Integrating these ideas, we found common keywords for solving the problems: human relations (with children’s guardians, children, boss, and colleagues, a place to share suffering), work conditions (salary, workforce, holidays, workload, and welfare benefits), and mental health. As such, the focus should be on a reward system, welfare benefits from the government or workplace, work conditions, human relations, and mental health. Additionally, preschool workers need places to share their suffering and learn to care for children and their guardians.

First, an analysis of work conditions suggested that a supportive work environment and better welfare benefits (e.g., flexible work options and paid maternal or paternal leave) and home telecommuting allowed them to continue working [8,9]. Similarly, efforts should be made to improve the conditions in both work-related and outside-work areas [10]. For example, Bhattacharya and Ramachandran reported that heavy workload-related pressures compel workers to leave work [11]; thus, supervisors should attempt to maintain the right workload balance [12].

Second, in terms of human relations, some studies have reported that challenging workplace relationships can lead to workplace stress across occupations and that providing counseling support for human relations can lead to better mental health [13,14,15,16,17]. Third, in terms of mental health, a survey indicated that a greater involvement in their work might reduce workers’ stress and increase job satisfaction.

Finally, another study suggested that better government implementation of stress management plans could benefit the workers’ psychosocial needs [18]. However, a randomized controlled trial found that nondirective social support focused on workers’ intrapsychic challenges was significantly associated with fewer health problems [19,20,21,22]. Furthermore, nondirective social support, which focuses on workers’ intrapsychic challenges, often improves positive health behavior, health outcomes, life satisfaction, self-esteem, hope, and optimism [20,21,23].

Hence, a better reward system (e.g., higher salary), welfare benefits (e.g., maternal leave) from the government or workplace, work conditions (e.g., less overtime work), human relations (e.g., compliments from the boss), and mental health (e.g., places to share suffering) affect their willingness to continue working. This study also has certain limitations. In particular, we recruited participants from one prefecture in Japan; therefore, the findings may not be generalizable across Japan or in other countries. Additionally, the study was limited in its ability to explore differences in responses across key demographic categories such as gender, age, experience in the field, etc., which must be considered in future research. Finally, future research should include participants from other prefectures and countries and attempt more specific questions suggested by the present research (e.g., letting the participants choose important workshop ideas from the given list). Despite these limitations, the findings provide meaningful new insights for addressing the shortage of preschool workers in childcare institutions.

## 5. Conclusions

The findings of this study suggest that the workshop ideas to solve job problems were related to children’s guardians, taking care of children, understanding children, place to share suffering, basic skills for work, mental health, and the way of working. In addition, the ideas of conditions most conducive to continuing working longer were related to employment conditions, workforce, workload, human relations, relationship with boss/seniors, work environment, welfare benefits, duty framework, taking care of children, taking care of children’s guardians, and mental conditions. As such, better reward systems, welfare benefits from the government or workplace, work conditions, human relations, and mental health can be key elements of counterplans to encourage preschool workers to continue working. In addition, these recommendations may assist in effectively addressing the high turnover rate among preschool workers in Japan.

## Figures and Tables

**Table 1 children-10-00032-t001:** Reasons for limiting willingness to continue employment to five years or less as a full-time preschool worker.

Reasons for Willingness to Leave Employment	*n* = 502 *n* (%)
Too much overtime work	258 (51)
Low salary	233 (46)
Human relations	144 (29)
Marriage	122 (24)
Differences in childcare/education policies between workplace and oneself	100 (20)
Difficulties in taking care of children’s guardians	90 (18)
Pregnancy/giving birth	77 (15)
Difficulties in taking care of children	74 (15)
Unsatisfactory welfare benefits	57 (11)
Working time/day problems	54 (11)
Inadequate levels of satisfaction at work	53 (11)
Other reasons	52 (10)
Health problems	48 (10)
Family problems	39 (8)
Changing jobs	41 (8)

The reasons with the highest percentages were “too much over time work”, followed by “low salary” and “human relations”.

**Table 2 children-10-00032-t002:** Categorized reasons for considering quitting a job as a full-time preschool worker.

No.	Reasons for Possibly Quitting the Job	*n* = 344 *n* (%)
1	Low salary	70 (20)
2	Too much overtime work/no payment for overtime	52 (15)
3	Too much workload	51 (14)
4	Human relations	50 (15)
5	Difficulties balancing family life and work	49 (14)
6	Too much take-out work	43 (13)
7	Relationships with directors/bosses/seniors	37 (11)
8	Work on holidays	34 (10)
9	Mental stress	31 (71)
10	Differences in childcare/education policies between workplace/others and oneself	28 (8)
11	Difficulties in taking care of children’s guardians	20 (6)
12	Framework of workplace	20 (6)
13	Working time/day problems	19 (6)
14	Health problems	19 (6)
15	Difficulties obtaining days off	18 (5)
16	Too much document work	16 (7)
17	Other reasons	16 (7)
18	Marriage	13 (4)
19	Power harassment	13 (4)
20	Difficulties in taking care of children	12 (3)
21	Inadequate levels of satisfaction at work	11 (3)
22	Lack of workers	11 (3)
23	Other challenges	11 (3)
24	Complaint for directors	10 (3)
25	Difficulties procuring childcare/education at work	9 (3)
26	No rest time	9 (3)
27	Contents of work	9 (3)
28	Pregnancy/giving birth	8 (2)
29	Weight of responsibility	7 (2)
30	Differences in concept of values with others	7 (2)
31	Unsatisfactory welfare benefits	6 (2)
32	Too many events at workplace	6 (2)
33	Not acknowledged by others	5 (1)
34	No confidence	5 (1)
35	No work engagement	5 (1)

The reasons with the highest percentages were “low salary”, followed by “too much overtime work/no payment for overtime” and “human relation”.

**Table 3 children-10-00032-t003:** Categorized contents and context of OT work performed by participating preschool workers in the last 30 days.

No.	Tasks Performed in OT Work in the Last 30 Days (*n* = 633)	OT Work Done at Workplace *n* (%)	OT Work Done at Home *n* (%)
1	Preparation for events	263 (42)	157 (25)
2	Administrative work	194 (31)	419 (66)
3	Extension of childcare/education	122 (19)	2 (0)
4	Preparation for childcare/education	113 (18)	53 (8)
5	Meeting	42 (7)	1 (0)
6	Preparation for educational tools/ teaching materials	37 (6)	54 (9)
7	Making teaching materials	30 (5)	85 (13)
8	General work	25 (4)	3 (0)
9	Taking care of guardians	24 (4)	0
10	Cleaning/decluttering	24 (4)	0
11	Others	17 (3)	9 (1)
12	Miscellaneous duties	16 (3)	3 (0)
13	Attendance for events	14 (2)	2 (0)
14	Workshop/research	10 (2)	6 (1)
15	Set up environment	8 (1)	1 (0)
16	Over-thinking	7 (1)	6 (1)
17	Cultivation/education for juniors	5 (1)	1 (0)
18	Playing a piano	4 (1)	16 (3)

OT: overtime. The contents at workplace with the highest percentage were “preparation for events”, followed by “administrative work”, and “extension of childcare/education.” The contents at home with the highest percentage were “administrative work”, followed by “preparation for events”, and “making teaching materials”.

## Data Availability

The datasets generated and analyzed during the current study are not publicly available due to a lack of participants’ agreement to put the data in public but are available from the corresponding author on reasonable request.

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
