# Peer review of "Why Do They Leave? The Counterplans to Continue Working among Preschool Workers in Japan: A Cross-Sectional Survey"

_children, 2022, doi:10.3390/children10010032_

Round 1

Reviewer 1 Report

I would like to thank you for the opportunity since I feel very fortunate to be able to review this article and I would like to congratulate the authors for this work. For me, as an educator specialized in the initial training of preschool teachers, this topic is very important and has a lot of value. I detail my suggestions below and in concluding my consideration.

This manuscript investigated the reasons why preschool/daycare/kindergarten teachers and workers leave employment and provide concrete proposals to reduce the turnover rate in these occupations.

Title: The title is concrete, representative and indicative of the problem investigated in the manuscript. As a suggestion, the title should provide information on where the research was conducted and provide information on the subject group.

Abstract: The abstract is clear and complies with the general rules for writing a good abstract. However, I would like to see a better description of the sample. This is the most important section of the paper since it will be read many more times than even the manuscript itself, so it needs the most attention. A brief note on the importance of the research is an excellent ending to a high-level abstract.

Introduction

As I mentioned, I find this research extremely important in contributing to the field of education. I do not disagree with the authors' justifications and read many very good and current arguments.  It is suggested to specify the ages covered by this preschool stage and to expand more information on the state of the art.

It is suggested to the authors that based on the stated objective they highlight the research questions that help to conduct the research and discussion based on the findings found in which the study variables, the study population, and the expected result appear.

Material and method.

It is suggested to include the type of design used and the type of methodology.

Instruments: It is suggested to better detail the instrument and include a model as an annex. 

Participants. This section should be better defined. In this section (participants), the characteristics of the sample should be included. There is no information on the sociodemographic data of the participants, characteristics, inclusion-exclusion criteria (extract them from the procedure section)... Was any type of sample calculation performed? It is suggested that a sample characterization table be included in the results section.

Results: The results are shown correctly and are easy to read. 

Discussion: It seems to me that a great job has been done in comparing the findings with other studies. Congratulations. It is suggested to include a section on practical and theoretical implications to evaluate the scope of the research.

Conclusions: They are clear and provide an answer to the stated objectives. 

Author Response

Response to the comments from the reviewers

We greatly appreciate the comments and suggestions provided by the reviewer, which were very valuable in helping us to further improve the quality of our manuscript. We have revised the manuscript as much as possible, in line with the reviewer’s suggestions. The revised parts are in red colored text. We hope that these corrections and revisions are satisfactory and that the revised version of our manuscript is now acceptable for publication in Children.

Response to the comments from reviewer 1

Title: The title is concrete, representative and indicative of the problem investigated in the manuscript. As a suggestion, the title should provide information on where the research was conducted and provide information on the subject group.

Response: Thank you for your kind suggestions regarding the Title. Following your comments, we have added the relevant information in the title (Page 1, Line 3).

Abstract: The abstract is clear and complies with the general rules for writing a good abstract. However, I would like to see a better description of the sample. This is the most important section of the paper since it will be read many more times than even the manuscript itself, so it needs the most attention. A brief note on the importance of the research is an excellent ending to a high-level abstract.

Response: Thank you for pointing this out. We have added the information in the Abstract section (Page 1, Lines 14–15).

Introduction

As I mentioned, I find this research extremely important in contributing to the field of education. I do not disagree with the authors' justifications and read many very good and current arguments. It is suggested to specify the ages covered by this preschool stage and to expand more information on the state of the art.

Response: We have added more information in the Introduction section as per your suggestion (Page 1, Lines 26–28).

It is suggested to the authors that based on the stated objective they highlight the research questions that help to conduct the research and discussion based on the findings found in which the study variables, the study population, and the expected result appear.

Response: We have added the expected result in the Introduction section (Page 2, Lines 52–53).

Material and method.

It is suggested to include the type of design used and the type of methodology.

Response: We have added the relevant information in the Materials and Methods section (Page 2, Line 64).

Instruments: It is suggested to better detail the instrument and include a model as an annex.

Response: In this study, we did not use any specific instruments and asked the participants to answer the counterplans questionnaire descriptively. We have added this explanation to the Materials and Methods section (Page 2, Lines 78–79).

Participants. This section should be better defined. In this section (participants), the characteristics of the sample should be included. There is no information on the sociodemographic data of the participants, characteristics, inclusion-exclusion criteria (extract them from the procedure section)... Was any type of sample calculation performed? It is suggested that a sample characterization table be included in the results section.

Response: We have explained our sampling method in the Methods section. A total of 551 responses were received from among the 1,002 contacted; we did not decide the exact total number of the sample and sociodemographic data was not received. The sampling methods were all based on previous research (Page 2, Lines 56–58).

Reviewer 2 Report

I realize that great work and time have been devoted to this paper. It has a lot of strengths, but I think that some changes should be recommended. 

Title: the title does not adequately reflect the content of the paper. Please, try to change it to better inform the readers about the relationships between the variables you test and also inform them about the quality of your sample.

Abstract:

Less information appears in the abstract. Maybe expanded by adding the most relevant findings. Please, take into account that the abstract is the unique part of your paper that most of the readers can read. Hence, more information would be better. Here, and in the rest of the paper, avoid using the word “impact”. Your methods are correlational, not experimental, and you could not affirm any causal relationship. 

Keywords: it is better to enlist your keywords alphabetically. Do not use keywords already captured in the title of the manuscript.

Introduction

The literature revision has some references that are too old. Besides citing some papers from 2002, you can consider some relevant papers on the topic RECENTLY published in other Journals. There are some Journals that suggest a high percentage of references published during the last five years. The introduction is too brief. The most relevant findings related to the constructs under study should be summarized.

Methodology

The Instruments or Questionnaires section needs more information. If you can, please inform me about previous studies where the same instrument has been used and the reliability obtained in that research.

Results

You only offer a descriptive analysis of frequencies and percentages, but it is not enough exploration of the data. Please, can you conduct a more refined analysis? For instance, SPSS offers the opportunity of correspondence analysis, which displays the frequency of any responses as a function of participants' categories (gender, type of school, age, etc.)

Discussion:

First of all, try to adjust your conclusions to the findings better. Or to say in other words, please try to justify more clearly the connection between your conclusions and your results.

Finally, a section related to limitations, future lines of investigation, and the principal contributions of the research could be attractive. Your paper has a lot of relevant implications for educators, psychologists, society, and policymakers, but you need to elaborate more on this topic.

Author Response

Response to the comments from the reviewers

We greatly appreciate the comments and suggestions provided by the reviewer, which were very valuable in helping us to further improve the quality of our manuscript. We have revised the manuscript as much as possible, in line with the reviewer’s suggestions. The revised parts are in red colored text. We hope that these corrections and revisions are satisfactory and that the revised version of our manuscript is now acceptable for publication in Children.

Response to the comments from reviewer 2

Title: the title does not adequately reflect the content of the paper. Please, try to change it to better inform the readers about the relationships between the variables you test and also inform them about the quality of your sample.

Response: Thank you for your kind suggestions regarding the title. Following your comments, we have changed it, appropriately. (Page 1, Line 2).

Abstract:

Less information appears in the abstract. Maybe expanded by adding the most relevant findings. Please, take into account that the abstract is the unique part of your paper that most of the readers can read. Hence, more information would be better. Here, and in the rest of the paper, avoid using the word “impact”. Your methods are correlational, not experimental, and you could not affirm any causal relationship.

Response: Based on your suggestion, we have changed the term “impact” to “correlate” throughout the manuscript, and added the most relevant findings in the Abstract (Page 1, Lines 20–21).

Keywords: it is better to enlist your keywords alphabetically. Do not use keywords already captured in the title of the manuscript.

Response: We have enlisted the keywords alphabetically (Page 1, Line 22).

Introduction

The literature revision has some references that are too old. Besides citing some papers from 2002, you can consider some relevant papers on the topic RECENTLY published in other Journals. There are some Journals that suggest a high percentage of references published during the last five years. The introduction is too brief. The most relevant findings related to the constructs under study should be summarized.

Response: Thank you for your comments. We have cited a recently published article and added more information in the Introduction section (Page 1, Lines 34–35).

Methodology

The Instruments or Questionnaires section needs more information. If you can, please inform me about previous studies where the same instrument has been used and the reliability obtained in that research.

Response: In this study, we did not use any specific instruments and asked the participants to answer the counterplans questionnaire descriptively. We have added this explanation to the Materials and Methods section (Page 2, Lines 78–79).

Results

You only offer a descriptive analysis of frequencies and percentages, but it is not enough exploration of the data. Please, can you conduct a more refined analysis? For instance, SPSS offers the opportunity of correspondence analysis, which displays the frequency of any responses as a function of participants' categories (gender, type of school, age, etc.)

In this study, we aimed to know the reality of preschool workers’ thoughts. Thus, it is difficult to analyze the survey information due to the lack of numerical data. Thus, we have explained this in the limitation section and will consider this in future research studies (Page 10, Lines 190–193).

Discussion:

First of all, try to adjust your conclusions to the findings better. Or to say in other words, please try to justify more clearly the connection between your conclusions and your results.

Response: We have justified the information more clearly in the Discussion section (Pages 10 imitation section and will consider this in future research studies (Page 10, Lines 190–11, Lines 203 imitation section and will consider this in future research studies (Page 10, Lines 197–203).

Finally, a section related to limitations, future lines of investigation, and the principal contributions of the research could be attractive. Your paper has a lot of relevant implications for educators, psychologists, society, and policymakers, but you need to elaborate more on this topic.

Response: We have added more information on limitations in the Discussion section (Page 1o, Lines 190–193).

Round 2

Reviewer 1 Report

I am very grateful to the authors for taking my considerations into account. I believe that the manuscript has improved sufficiently to be published in children. Thank you very much

Author Response

Response to the comments from the reviewer1

I am very grateful to the authors for taking my considerations into account. I believe that the manuscript has improved sufficiently to be published in children. Thank you very much

We greatly appreciate the comments and suggestions provided by the reviewer. We are also grateful that you were satisfied with the revised version of the manuscript. We hope that the revised version of our manuscript is now acceptable for publication in Children.
